# Change in confidence in public health entities among US adults between 2020–2024

**Hannah Melchinger**, **Saad B. Omer**, **Amyn A. Malik***

Peter O'Donnell Jr. School of Public Health, UT Southwestern Medical Center, Dallas, Texas, United States of America

* amyn.malik@utsouthwestern.edu

## Abstract

The COVID-19 pandemic introduced significant uncertainty among Americans, and eroded trust in US health entities and institutions. This study describes a longitudinal assessment of Americans' change in confidence in these public health entities from 2020 to 2024. We conducted four surveys assessing confidence in public health entities among US adults conducted between February 2020 and October 2024. Surveys were hosted by CloudResearch and administered to a representative sample of US adults (18+). Data was weighted using age, gender, and race estimations sourced from the American Community Survey (ACS) 2022. From February 2020 to June 2022, Americans' confidence declined for all entities asked about, most notably the Centers for Disease Control and Prevention (CDC, 26%). Despite slight increases between 2022 and 2024, levels of confidence have not recovered to pre-pandemic values. In contrast, Americans' confidence in their own doctor and their local health department increased from 2022 to 2024 (5% and 19%, respectively), as did confidence in the White House (7%). Our study saw a marked decrease in confidence in public health institutions among US adults, a finding which has implications for the effectiveness of public health communication. Our results highlight the potential for local entities, including personal doctors and local health departments, to leverage the existing trust between them and their public to provide essential public health information and address low confidence in national entities.

## Introduction

The COVID-19 pandemic introduced an unprecedented level of uncertainty among the United States public. Notably, survey data has shown that confidence in public health entities among adults in the United States has decreased significantly since 2020. Here we describe one of the first academic studies to longitudinally assess

**Data availability statement:** Data can be accessed here: https://github.com/OmerResearchGroup/Change-in-confidence-in-public-health-entities-among-US-adults-between-2020-2024.

**Funding:** The authors received no specific funding for this work.

**Competing interests:** The authors have declared that no competing interests exist.

changes in confidence in health entities among US adults between 2020 and 2024.

## Methods

### Survey development and dissemination

Between February 2020 and October 2024, we conducted four discrete, cross-sectional surveys assessing confidence in public health entities among US adults [1–4]. Surveys were constructed using Qualtrics (Qualtrics, Provo, UT) and administered by CloudResearch, an online survey company, to a census-matched sample of US adults (18+). A representative sample was recruited from CloudResearch's pre-existing survey panels, which consist of individuals recruited by word-of-mouth and advertising to take surveys. Respondents were recruited according to demographic quotas proportional to the U.S. Census; within each quota, respondents were selected randomly [5]. Participants who completed the survey received a nominal incentive (~2 USD), which they could redeem in several ways, for example as cash, subscription credits, or donations. Distinct representative study populations were recruited from the same panels for all studies. Survey 1 was conducted from February 4–7, 2020; Survey 2 from May 6–7, 2020; Survey 3 from June 8–13, 2022; and Survey 4 from September 20–27, 2024. Sample sizes for Surveys 1–4 were 718, 672, 856, and 828, respectively.

Surveys 1 and 2 asked questions about to the COVID-19 pandemic, while Surveys 3 and 4 asked questions around the 2022 and 2024 mpox outbreaks. In addition to these disease-specific questions, we asked participants a series of questions about their perceptions of public health entities, including who they thought should lead the US response to infectious disease outbreaks, and how they would rate their confidence in different public health entities.

### Analysis

Survey results were analyzed using STATA SE 18.5. Survey weights were calculated based on age, gender, and race estimations sourced from the American Community Survey (ACS) 2022. Confidence was measured on a scale of 1–5, with ratings of 4 and 5 categorized as "high confidence." Statistical significance was defined as $p < 0.05$. Means, proportions, and 95% confidence intervals (CI) were calculated and adjusted for survey weights. Data from Surveys 1–3 were analyzed in October 2024, along with data from Survey 4; no personal respondent data was accessed or analyzed.

### Ethics statement

Approval for Survey 1 (2000027402), Survey 2 (2000027891), and Survey 3 (2000032980) was granted by the Institutional Review Board (IRB) at Yale University, and by the IRB at University of Texas Southwestern Medical Center for Survey 4 (STU-2024–0933). Respondents were advised at the start of the survey that by continuing with the questionnaire, they were consenting to study participation.

Participation was voluntary and all survey results were kept confidential. No personal data was collected from respondents; survey metadata, included IP address, device type, and location, were not stored or used.

## Results

From February 2020 to October 2024, there has been a significant decline in mean confidence among US adults in health institutions including the Centers for Disease Control and Prevention (CDC), National Institutes of Health (NIH), the Department of Health and Human Services (DHHS), state health departments, and professional medical organizations (Fig 1). Between February and May 2020, the percentage of respondents reporting high confidence in the CDC decreased from 82% to 68%, before dropping to 56% in 2022 (Fig 2). Between 2022 and 2024, confidence in the CDC increased to 60%; however, the change was not statistically significant (S1 Table). Similarly, confidence in the NIH, DHHS, state health departments, and professional medical organizations dropped by 25%, 13%, 16%, and 26%, respectively, from February 2020 to June 2022. Despite slight increases between June 2022 and October 2024, confidence in all institutions remained significantly lower than in February 2020.

Despite decreasing from February 2020 to June 2022, respondents' confidence in their own doctor and local health departments increased significantly between 2022 and 2024 by 5% and 19%, respectively. Between February 2020 and 2024, confidence in physicians decreased overall by only 6%, while local health departments increased overall by 6%. Similarly, confidence in the White House increased by 10% from February 2020 (29%) to October 2024 (39%). Overall confidence in the Food and Drug Administration (FDA) remained largely unchanged.

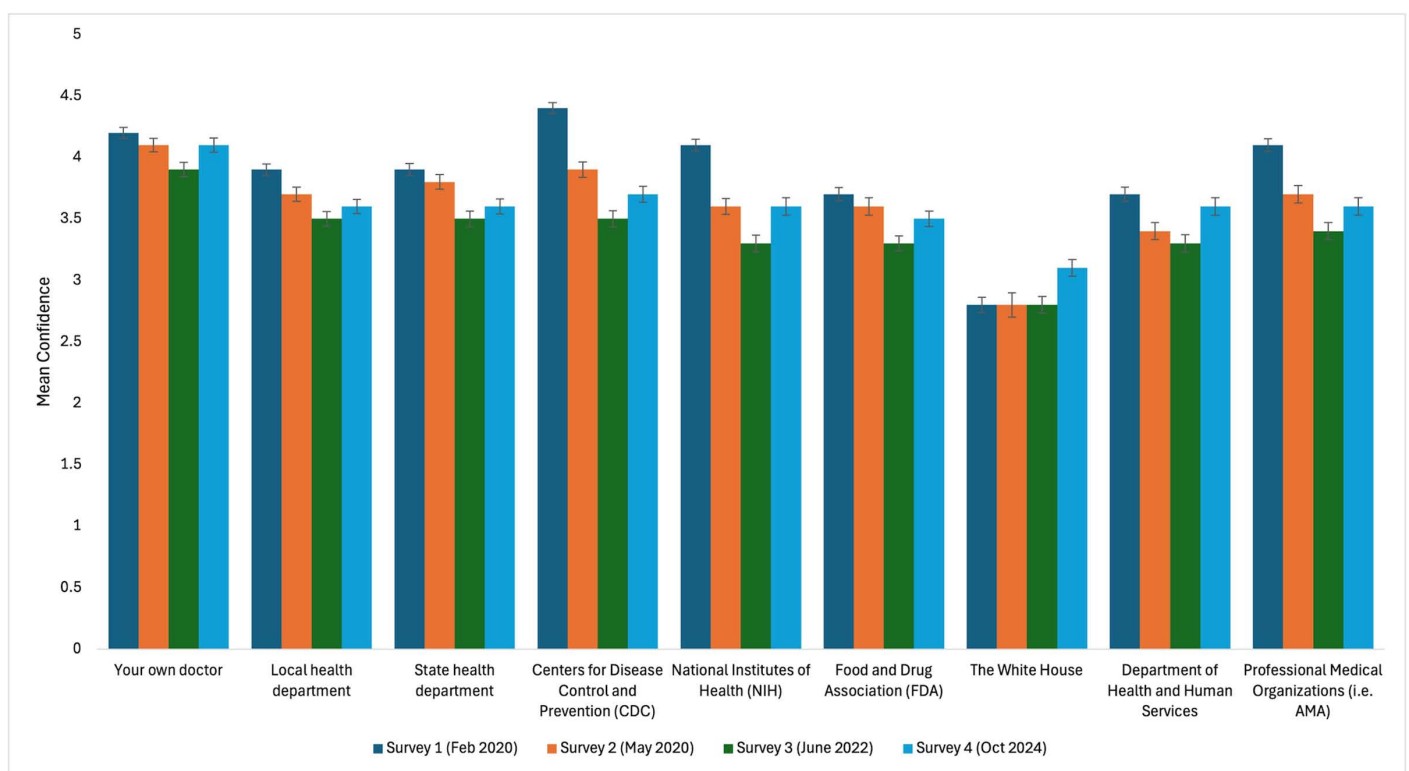

**Fig 1. Mean confidence in public health organizations among US adults.**

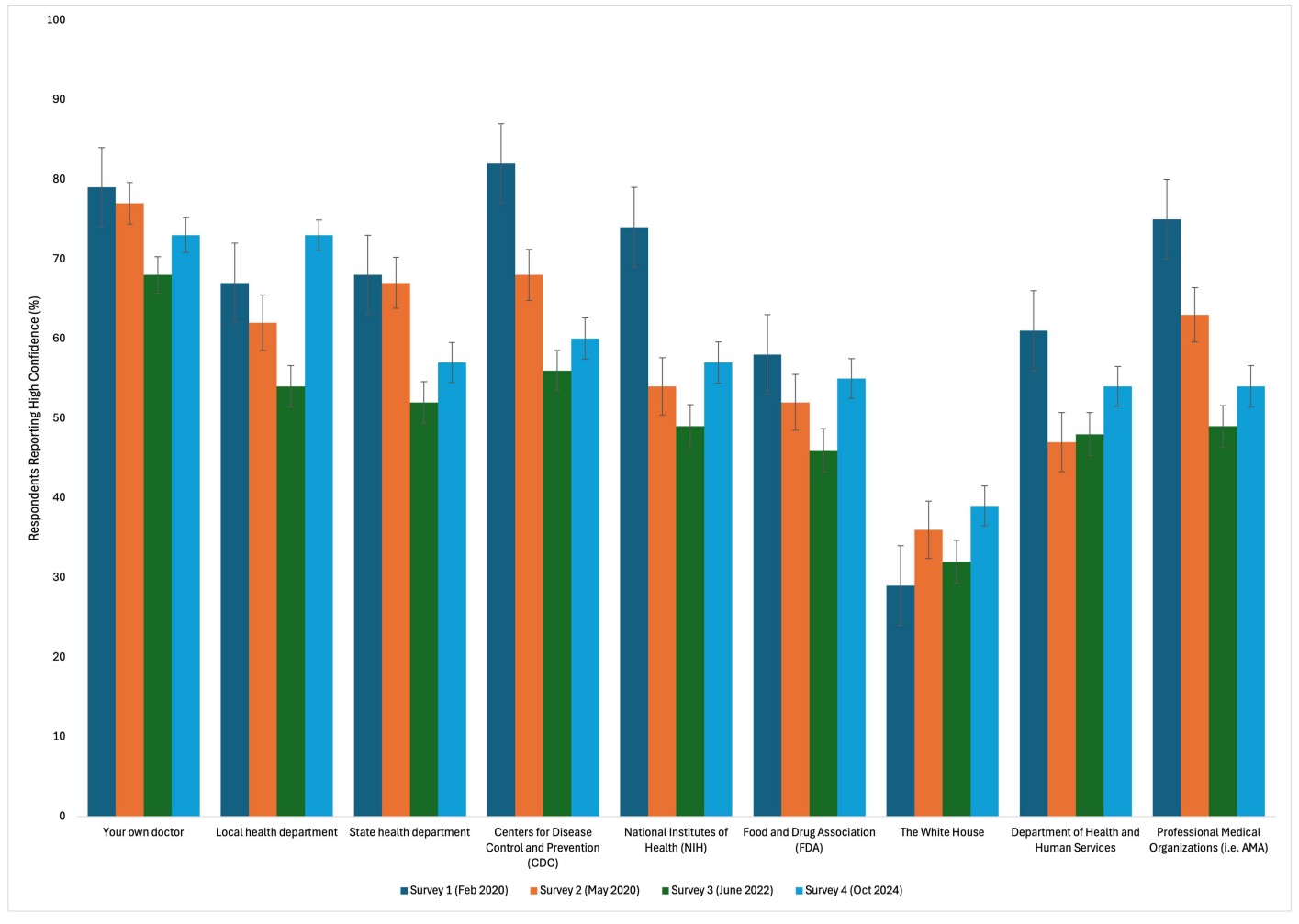

**Fig 2. Percentage of US adults reporting high confidence in public health organizations.**

## Discussion

Public health institutions like the CDC and NIH provide essential communication and resources to the American public, particularly during health emergencies. Our study documents a marked decrease in trust of public health entities among Americans, especially in premier institutions like the CDC and NIH. Several factors have been shown to contribute to this lack of confidence, including misinformation about healthcare and public health services, and socio-economic challenges caused by the pandemic [6]. Recent studies have suggested that behavioral interventions – especially those which use trusted messengers to communicate with less confident communities – may play a role in reestablishing trust in public health institutions and interventions [7]. Importantly, this study highlights the potential of local health entities, including physicians and local health departments, to act as those messengers and effectively communicate about public health. Local health authorities should be aware of this influence when they interact with patients and public, particularly when discussing complex health topics like COVID-19 vaccination [8]. The findings highlight the urgent need to address American's confidence in US health institutions through evidence-based interventions.

## Supporting information

**S1 Table. Change in weighted percentage reporting high confidence (2020–2024).**
(DOCX)

## Author contributions

**Conceptualization:** Hannah Melchinger, Saad B. Omer, Amyn A. Malik.

**Formal analysis:** Hannah Melchinger, Amyn A. Malik.

**Methodology:** Hannah Melchinger.

**Project administration:** Hannah Melchinger, Amyn A. Malik.

**Supervision:** Saad B. Omer, Amyn A. Malik.

**Writing – original draft:** Hannah Melchinger.

**Writing – review & editing:** Hannah Melchinger, Saad B. Omer, Amyn A. Malik.

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
