## [Decision Letter · Decision Letter 0]

7 May 2025

PGPH-D-25-00251

Change in confidence in public health entities among US adults between 2020-2024

Dear Dr. Malik,

Thank you for submitting your manuscript to PLOS Global Public Health. After careful consideration, we feel that it has merit but does not fully meet PLOS Global Public Health’s publication criteria as it currently stands. Therefore, we invite you to submit a revised version of the manuscript that addresses the points raised during the review process.

We look forward to receiving your revised manuscript.

Kind regards,

Nafis Faizi, MD, MPH

Academic Editor

Journal Requirements:

1. In the online submission form, you indicated that De-identified data can be provided upon reasonable request. 

3. Uploaded as supplementary information.

2. We notice that your supplementary tables are included in the manuscript file. Please remove them and upload them with the file type 'Supporting Information'. Please ensure that each Supporting Information file has a legend listed in the manuscript after the references list.

Additional Editor Comments (if provided):

Reviewers' comments:

Reviewer's Responses to Questions

**Comments to the Author**

1. Does this manuscript meet PLOS Global Public Health’s publication criteria ? Is the manuscript technically sound, and do the data support the conclusions? The manuscript must describe methodologically and ethically rigorous research with conclusions that are appropriately drawn based on the data presented.

Reviewer #1: Yes

Reviewer #2: Yes

2. Has the statistical analysis been performed appropriately and rigorously?

Reviewer #1: Yes

Reviewer #2: I don't know

3. Have the authors made all data underlying the findings in their manuscript fully available (please refer to the Data Availability Statement at the start of the manuscript PDF file)?

Reviewer #1: Yes

Reviewer #2: Yes

4. Is the manuscript presented in an intelligible fashion and written in standard English?

Reviewer #1: Yes

Reviewer #2: Yes

5. Review Comments to the Author

Reviewer #1: Your methods section is well-organized and provides a clear overview of survey development, analysis, and ethical considerations. However, there are a few areas where clarity, consistency, and detail can be improved.

Strengths: the sections flow smoothly. You specify the use of CloudResearch and Qualtrics, as well as the survey timeframe and incentives. Appropriate statistical analysis methods – Use of survey weighting, confidence intervals, and p-values is correctly described. IRB approvals and ethical considerations are well-documented.

Areas of improvement:

Clarify survey design (cross-sectional, longitudinal, or sampling?).

The sentence "Between February 2020 and October 2024, we conducted four surveys assessing confidence in public health entities..." implies a continuous process, but surveys were conducted at four distinct time points (not continuously). Specify whether the surveys were discrete cross-sectional assessments or an ongoing study.

Standardize date formatting for consistency.

"Participants were recruited from February 4-7, 2020, for Survey 1, from May 6-7, 2020, for Survey 2, from June 8-13, 2022, for Survey 3, and from September 20-27, 2024, for Survey 4." The mix of different prepositions and punctuation makes this harder to read. Consider "Survey 1 was conducted from February 4–7, 2020; Survey 2 from May 6–7, 2020; Survey 3 from June 8–13, 2022; and Survey 4 from September 20–27, 2024."

Were participants randomly selected? Stratified? Quota-based?

"Survey weights were calculated based on age, gender, and race estimations sourced from the American Community Survey (ACS) 2022." What about education level, geographic region, or income? Why were they excluded?

Strengthen survey weighting and statistical analysis description.

"Significance was determined as anything with a p-value less than an alpha of 0.05." sounds too informal—use "statistical significance was defined as p <0.05."

Improve ethical statement to ensure clarity on data privacy & consent.

"Participants were advised at the start of the survey that by continuing with the questionnaire, they were consenting to participating in the study." Consider "consenting to participate."

Address potential metadata collection concerns for better confidentiality assurances.

The statement "No personal data was collected from respondents." is good, but does it include IP addresses, cookies, or metadata? Qualtrics automatically collect metadata (e.g., location, device type). If this data was not stored or used, clarify this to strengthen data privacy assurances.

Reviewer #2: The topic is interesting. The manuscript has a good flow and is written in brief and an interesting way. I have a few suggestions to consider.

Was the study population same for all four surveys? While the method section describe the participants were recruited from pre-existing panels in the CloudResearch survey system, but do not explicitly state whether the same individuals participated in all four surveys or if each survey comprised a new, independent sample drawn from the same pool.

What was the sample size of all four surveys?

The IRB permission from two different universities also create slight confusion. For the first three survey (2020 and 2022) Institutional Review Board permission taken from Yale University while for 4th survey (2024) taken from University of Texas. These two university are located in different states, and their residents may differ in perception regards public entities. Consider highlighting it in limitation section, if the survey population is different in these four surveys.

In methods section last para – ‘Among disease-specific questions’, can you explain what do you mean by this, I could not see its relevance in the current study. Kindly rewrite the sentence for clarity (confidence in public health entities repeated twice in same sentence)

6. PLOS authors have the option to publish the peer review history of their article (what does this mean? ). If published, this will include your full peer review and any attached files.

**Do you want your identity to be public for this peer review?** For information about this choice, including consent withdrawal, please see our Privacy Policy .

Reviewer #1: **Yes: ** Obianuju Genevieve Aguolu

Reviewer #2: No

---

## [Editor Report · Decision Letter 1]

4 Feb 2025

Change in confidence in public health entities among US adults between 2020-2024

PGPH-D-25-00251R1

Dear Dr. Malik,

We are pleased to inform you that your manuscript 'Change in confidence in public health entities among US adults between 2020-2024' has been provisionally accepted for publication in PLOS Global Public Health.

Best regards,

Nafis Faizi, MD, MPH

Academic Editor